# ConvNet vs Transformer, Supervised vs CLIP: Beyond ImageNet Accuracy

## Abstract

Modern computer vision offers a great variety of models to practitioners, and selecting a model from multiple options for specific applications can be challenging. Conventionally, competing model architectures and training protocols are compared by their ImageNet accuracy. However, this single metric does not fully capture performance nuances critical for specialized tasks. In this work, we conduct an in-depth comparative analysis of model behaviors beyond ImageNet accuracy for four leading models: ConvNeXt and Vision Transformer (ViT), across supervised and CLIP training objectives. Although selected models have similar ImageNet accuracies and compute requirements, we find that they differ in many other aspects — types of mistakes, output calibration, transferability, and feature invariance, among others. This diversity in model characteristics, not captured by traditional metrics, provides insights for better model selection to meet specific goals. Our research also highlights the need for more nuanced analysis when choosing among models.

## 1 Introduction

The model landscape in computer vision has become increasingly complex. From early ConvNets (LeCun et al., 1998) to recent advances in Vision Transformers (Dosovitskiy et al., 2020), the variety of models available has expanded significantly. Similarly, training paradigms evolved from supervised training on ImageNet (Deng et al., 2009) to self-supervised learning (Chen et al., 2020; He et al., 2020) and image-text pair training methods like CLIP (Radford et al., 2021). While indicative of progress, this explosion of choices poses a significant challenge for practitioners: how to select a model that suits their purposes.

Traditionally, ImageNet accuracy has served as the primary metric for evaluating model performance. It has driven tremendous progress in the past 15 years, igniting the deep learning revolution (Krizhevsky et al., 2012). However, this metric is becoming increasingly insufficient. While ImageNet is useful to measure a model's general capability, it does not capture the nuanced model differences arising from varying architectures, training paradigms, and data – models with different properties may appear similar when judged solely based on ImageNet accuracy (Fig. 1). This limitation becomes more pronounced as models start to fit the idiosyncrasies of ImageNet with saturated accuracies (Beyer et al., 2020).

A particularly noteworthy example is CLIP. Despite having a similar ImageNet accuracy as a ResNet (He et al., 2016), CLIP's vision encoder has been shown to exhibit significantly better robustness and transferability. This has sparked research

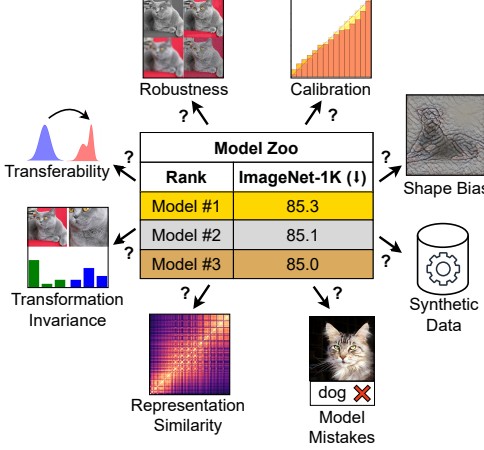

Figure 1: Models are sometimes compared only by their ImageNet accuracy, ignoring many other aspects of their behaviors.

that builds upon the unique strengths of CLIP (Ramesh et al., 2022; Luo et al., 2022; Wortsman et al., 2022), greatly illustrating the limitations of focusing solely on ImageNet accuracy.

The growing integration of vision models into production systems calls for a deep understanding of their behaviors. Conventional metrics do not fully capture models' ability to handle real-world vision challenges like varying camera poses, lighting conditions, or occlusions. Such a one-dimensional view becomes especially problematic in high-stakes applications (Gao et al., 2019). For example, a self-driving system might perform well in standard settings but could falter when exposed to unusual road conditions. In this case, only focusing on conventional accuracy metrics could result in serious consequences.

To bridge this gap, we conduct an in-depth exploration focusing on model behaviors beyond standard metrics. We analyze four leading models in computer vision: ConvNeXt (Liu et al., 2022) (a representative modern ConvNet) and Vision Transformer (ViT), with supervised and CLIP training. The selected models are similar in size and show nearly identical accuracy on ImageNet-1K within each training method, ensuring a fair comparison. Our study delves into a wide array of model characteristics, such as types of prediction errors, generalization capabilities, invariances of the learned representations, calibration performance, and many others.

We discover substantial variations in model behaviors among different architectures and training methods. For example, we show that CLIP models make fewer classification errors relative to their ImageNet performance and have great transferability. However, supervised models are better calibrated and generally superior on ImageNet robustness benchmarks. ConvNeXt models have advantage on synthetic data but more texture-biased than ViT, suggesting that architecture is still a crucial factor. Based on these findings, it becomes clear that different models can be useful under different metrics. We hope our analysis can help practitioners select models. Our research emphasizes the need for nuanced evaluation metrics for accurate, context-specific model selection.

## 2 MODELS

Our goal is to investigate the influence of architecture and training paradigms on model characteristics and capabilities. ViT and ConvNeXt are two representative architectures from the Transformer and ConvNet model families, respectively. Supervised models continue to show state-of-the-art performance in computer vision (Dehghani et al., 2023). CLIP-based models, on the other hand, excel in generalization and transferability and offer intriguing representational properties that connect vision and language. Self-supervised models (He et al., 2022; Woo et al., 2023) are not included in our study as they showed performance similar to supervised models in preliminary tests.

For supervised models, we use DeiT3-Base/16 (Touvron et al., 2022), which shares the same architecture as ViT-Base/16 with an improved training setting, and ConvNeXt-Base (Liu et al., 2022). For CLIP models, we use vision encoders ViT-Base/16 and ConvNeXt-Base from OpenCLIP (Ilharco et al., 2021). These models have the same size and compute requirement and are publicly available. Furthermore, the models have similar ImageNet-1K validation accuracies within their respective training paradigms and are trained on $224 \times 224$ images. A detailed comparison of selected models is presented in Table 1.

| Model | Architecture | Pretraining | Finetuning | Paradigm | FLOPs | #Param | INet-1K % |
|---|---|---|---|---|---|---|---|
| ViT-sup | DeiT3-B/16 | ImageNet-21K | ImageNet-1K | supervised | 17.5G | 87M | 83.7 |
| ConvNeXt-sup | ConvNeXt-B | ImageNet-21K | ImageNet-1K | supervised | 15.4G | 89M | 83.7 |
| ViT-clip | ViT-B/16 | LAION-400M | — | CLIP | 17.5G | 87M | 67.0 |
| ConvNeXt-clip | ConvNeXt-B | LAION-400M | — | CLIP | 15.4G | 89M | 66.3 |

Table 1: Model configurations in our analysis.

## 3 ANALYSIS

**1. Model Mistakes**

In image classification, a model mistake is an incorrect label assignment, such as misclassifying a cat as a dog. Simply identifying mistaken object classes might not offer actionable insights for

model improvement. The key aspect, therefore, is finding the specific reasons for these mistakes. For instance, some models may be particularly sensitive to certain aspects of the data distribution, like texture variations. In this case, a model might consistently make mistakes when the texture of the object differs from what it has been trained on. Identifying mistake types allows for targeted data collection and retraining, offering advantages over a black-box approach.

The ImageNet-X dataset (Idrissi et al., 2022) offers detailed human annotations for 16 factors of variation, such as pose and lighting. This allows a targeted analysis of models' mistake types. The annotations enable measuring model error ratios for each factor independently: error ratio(factor) $= \frac{1-\text{accuracy(factor)}}{1-\text{accuracy(overall)}}$, where accuracy(overall) is the overall ImageNet-1K validation accuracy, and accuracy(factor) is the accuracy on all the images where the given factor was highlighted. This metric measures the model performance on a given factor relative to its overall performance. Smaller error ratio values indicate better performance, as they suggest higher accuracy for the specified factor.

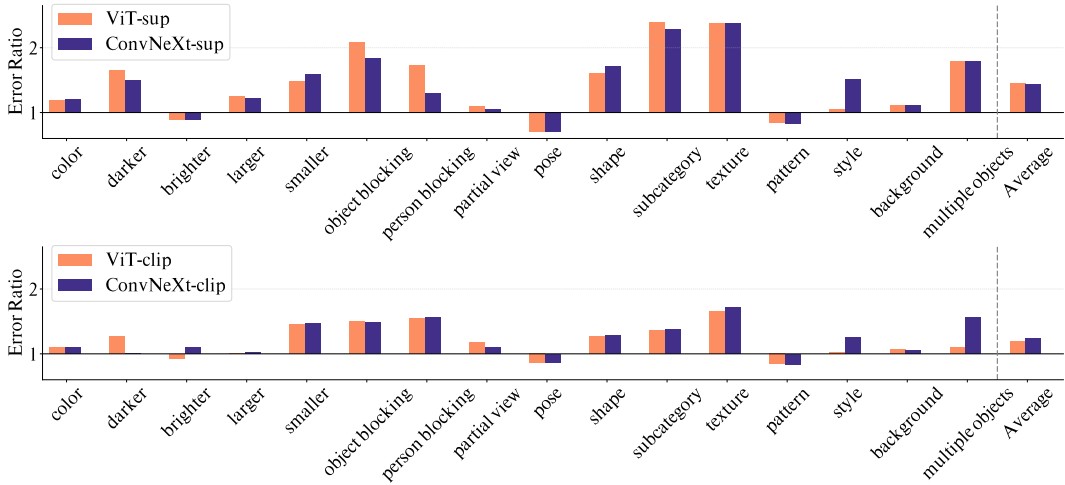

Figure 2: ImageNet-X results for each model mistake factor. Lower is better. CLIP models have smaller error ratios.

Results on ImageNet-X for our four selected models are presented in Fig. 2. The results show that CLIP models have a smaller error ratio, indicating a significant advantage. However, it's important to note that the error ratio is relative to overall ImageNet accuracy, where a significant 16% gap exists between supervised and CLIP zero-shot models. CLIP models are much more robust towards shape, subcategory, texture, object blocking, and darker factors. The key reason for the difference between CLIP and supervised models is likely the more diverse training data used for CLIP.

For CLIP models, there are three factors with dissimilar performance between ConvNeXt and ViT: multiple objects, style, and darker. For the first two, the ConvNeXt has a higher error ratio, while for the latter, it has an advantage over ViT. For supervised models, the performance only diverges for style and person blocking. Except for these factors, models largely have similar error ratios. The six factors for which all the models have a high error ratio are smaller, object blocking, person blocking, shape, subcategory, and texture. High error ratio factors usually involve complex visual scenarios, which helps to explain why models often make mistakes in these situations. For instance, in occlusion, the model often misclassifies due to focusing on the visible, obscuring object.

## 2. Synthetic Data

After exploring model mistakes on natural images, we turn our attention to synthetic ones. Unlike human-annotated data, synthetic datasets allow precise control over factors like camera angles, object positioning, and textures.

PUG-ImageNet (Bordes et al., 2023) is a synthetic dataset of photorealistic images of ImageNet classes that also provides labels for a set of factors. The images are generated using a software engine that allows systematically varying factors like pose, size, texture, lighting, and background

for each object. PUG-ImageNet facilitates in-depth analysis of model performance on synthetic data, an important area given that training models on synthetic data has emerged as a promising research avenue (Tian et al., 2023).

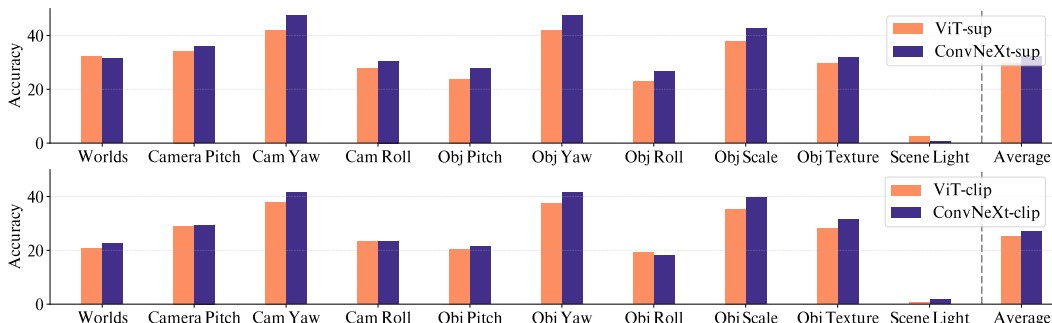

Figure 3: Results on PUG-ImageNet for each model mistake factor. Higher is better. ConvNeXt is better on almost every factor across both supervised and CLIP.

We provide accuracy results for ten different factors in PUG-ImageNet and their average in Fig. 3. Intriguingly, ConvNeXt outperforms ViT on PUG-ImageNet for nearly all factors except Scene Light, for which all models perform poorly. This suggests: ConvNeXt is better than ViT on synthetic data. For CLIP models, the gap between ConvNeXt and ViT is slightly smaller than that for supervised models, and they generally have lower accuracy compared to supervised models. This is likely related to their inferior accuracy on the original ImageNet.

## 3. Model Calibration

Calibration is a metric that quantifies the reliability of a model's predicted confidence levels (Guo et al., 2017). A model's confidence for a prediction is defined as the max probability among all classes in its output distribution. We are interested in determining whether the model is overly confident or too uncertain in its predictions. For instance, if the network deems a set of predictions to be 80% confident, does the actual accuracy hover around 80%?

Expected Calibration Error (ECE) measures the rate of calibration. To calculate ECE, predictions first need to be separated into the $M$ bins $B_1, \ldots, B_M$ based on their confidence. For instance, one bin can include all the predictions with confidence between 50% and 60% and so on. Each bin's confidence and accuracy are calculated as the average confidence and accuracy of predictions in $B_i$, represented as $\mathrm{conf}(B_i)$ and $\mathrm{acc}(B_i)$. Then, ECE can be defined as:

$$\mathrm{ECE} = \sum_i^M \frac{|B_i|}{n} |\mathrm{acc}(B_i) - \mathrm{conf}(B_i)|, \tag{1}$$

where $|B_i|$ is the size of the $i$-th bin.

Model calibration is also often assessed through visualizations including reliability diagrams and confidence histograms. Reliability diagrams plot the predicted confidence against accuracy; a well-calibrated model would show a graph where points closely align with the diagonal. Confidence histograms display how often different confidence levels occur in the model's predictions.

For a balanced evaluation, we present calibration metrics on two different datasets: ImageNet-1K for in-distribution data and ImageNet-R for out-of-distribution data. We selected ImageNet-R as the out-of-distribution dataset because CLIP models exhibit higher accuracy on it than supervised models. In all our experiments, we divide the data into $M = 15$ bins. We plot confidence histograms, reliability diagrams, and ECE scores in Fig. 4.

In Fig. 4, we observe that CLIP models have bars consistently below the diagonal in reliability diagrams and a notably high last bar in the confidence histogram, signaling overconfidence in both in-distribution and out-of-distribution data. Although Minderer et al. (2021) attributes calibration performance mainly to architecture, our results suggest otherwise: higher ECE scores in CLIP mod-

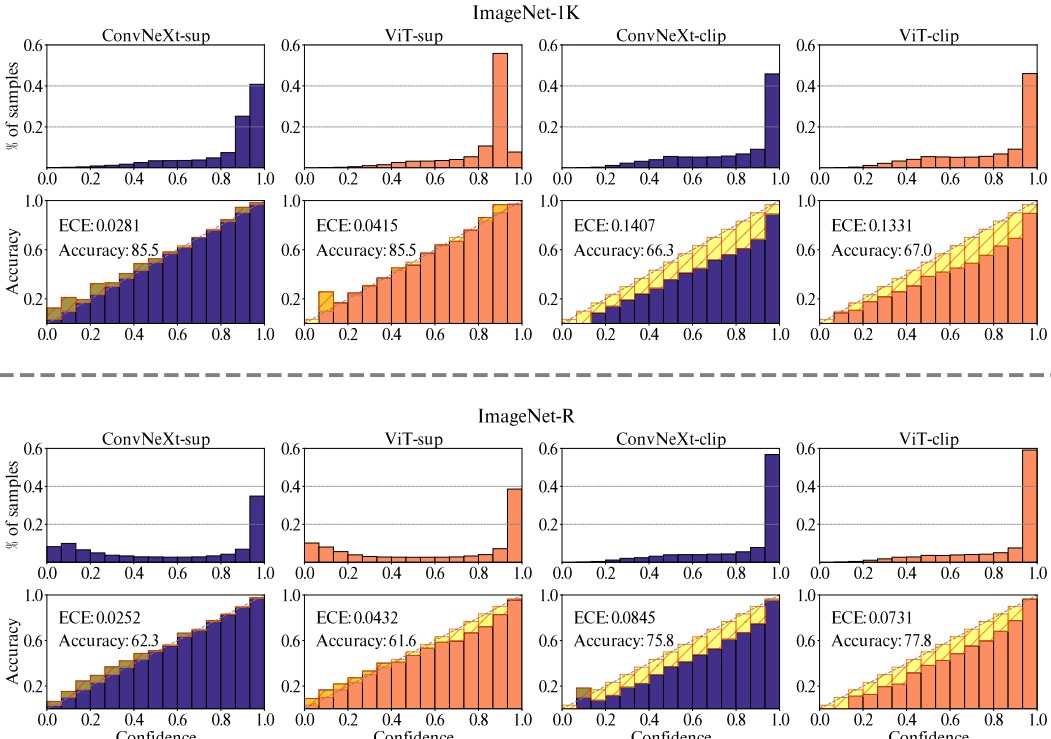

Figure 4: Confidence histograms (1st and 3rd row), reliability diagrams (2nd and 4th row), and ECE metric on ImageNet-1K (top part) and ImageNet-R (bottom part). Supervised models have lower ECE in both cases. ConvNeXt is competitive to ViT.

els, despite superior accuracy on ImageNet-R, indicate that training data and objectives could be more influential factors.

In the lower section of Fig. 4 related to ImageNet-R, we note that supervised models exhibit a higher density in the lower confidence intervals of the confidence histograms (3rd row). Additionally, these models show elevated accuracy levels in the initial bins of the reliability diagrams (4th row). These findings suggest that supervised models tend to be slightly underconfident.

Contrary to Minderer et al. (2021), which find that ViTs are better calibrated than ConvNets, our experiments show that supervised ConvNeXt have lower ECE score. This discrepancy might be because Minderer et al. (2021) focused on older ConvNet architectures, such as ResNet, while we use a more modern one. For CLIP models, we find that ViT is only slightly better than ConvNeXt.

## 4. Shape / Texture Bias

Humans can generally learn high-level cues for visual recognition, but neural networks often rely on more brittle shortcut features (Geirhos et al., 2020). The study of shape-texture bias (Geirhos et al., 2018) serves to highlight this phenomenon by examining model behavior on cue-conflict images, which contain a shape from one class superimposed with the texture from another. Two key metrics are introduced to quantify this bias: the shape and the texture fractions. The shape fraction calculates the proportion of decisions leaning towards the class represented by the shape, while the texture fraction accounts for the proportion favoring the class represented by the texture. These metrics reveal whether the classifier favors shape or texture when they conflict.

The study in Geirhos et al. (2018) demonstrates that convolutional networks have a strong bias towards texture, as opposed to shape, which differs from human behavior. In subsequent work, Naseer et al. (2021) concluded that ViT is less biased towards the texture than ConvNet by comparing the first generation of DeiT-S (Touvron et al., 2021) and ResNet-50. Notably, scaling large Transformer models has led to shape biases comparable to human level (Dehghani et al., 2023).

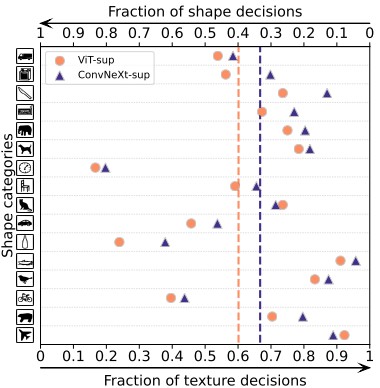 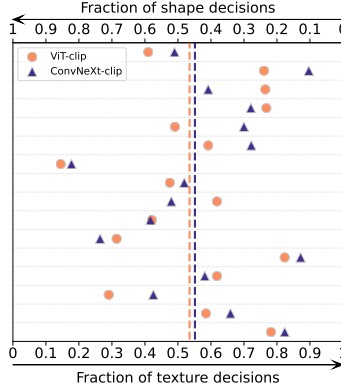

Figure 5: Fraction of shape vs texture decisions. CLIP and ViT models show a higher shape bias.

We evaluate shape-texture bias in our models using cue-conflict images and display the findings in Fig. 5. Dashed lines represent average shape bias aggregated over all the categories, while individual markers depict shape bias for the particular class.

In Fig. 5, we can observe that ViTs exhibit stronger shape bias than ConvNeXts for both supervised and CLIP models. This is possibly because ConvNeXt is more inclined to learn local features related to textures due to the local nature of convolution operations. However, the gap between ViT and ConvNeXt is much smaller for CLIP-based models. Notably, the shape bias in CLIP models improved by 7% and 12% for both architectures, prompting questions about the benefits of further scaling the training data. In Dehghani et al. (2023), it has been shown that a 22B parameter ViT model can achieve 87% shape bias. In our analysis, the highest result for ViT CLIP is 46.4%, suggesting that the model size might also play an important role.

## 5. Robustness

A model may excel on data from its training distribution but struggle to generalize to a distribution shift (Recht et al., 2019). These shifts can arise from natural perturbations such as atmospheric conditions (e.g., fog, rain), camera noise, or variations in object location and orientation. Model robustness quantifies a model's capability to adapt to changes in data distributions. A robust model should maintain high accuracy with these perturbations. This is particularly important for applications where reliability is a primary concern.

We evaluate the robustness on several datasets that feature many different types of natural variations and corruptions: ImageNet-V2 (Recht et al., 2019), ImageNet-A (Hendrycks et al., 2021b), ImageNet-C (Hendrycks & Dietterich, 2019), ImageNet-R (Hendrycks et al., 2021a), ImageNet-Sketch (Wang et al., 2019), ImageNet-Real (Beyer et al., 2020) and ImageNet-Hard (Taesiri et al.). We also provide ImageNet-1K validation accuracy for reference (INet-Val). We present the result plot in Fig. 6 (top half). We additionally provide full results in table format in the Appendix.

In Fig. 6, we can see that ViT and ConvNeXt, on average, exhibit similar performance across both supervised and CLIP. Supervised models perform better than CLIP on most datasets except ImageNet-R and ImageNet-Sketch. CLIP models' success on ImageNet-R and ImageNet-Sketch suggests they handle abstract or creative visuals better than supervised models. The advantage of supervised models is likely related to the fact that all robustness datasets share the same set of classes as the original ImageNet-1K, on which the supervised models were finetuned.

## 6. Transferability

The transfer learning performance of a model indicates its ability to adapt to new tasks and datasets beyond its original training domain (Kolesnikov et al., 2020). Good transferability allows for rapid finetuning with minimal additional effort, making it easier to scale the model to a wide range of real-world applications. The ability of a model to adapt to these shifts without significant degradation in performance serves as a valuable metric for its utility and generalization capabilities.

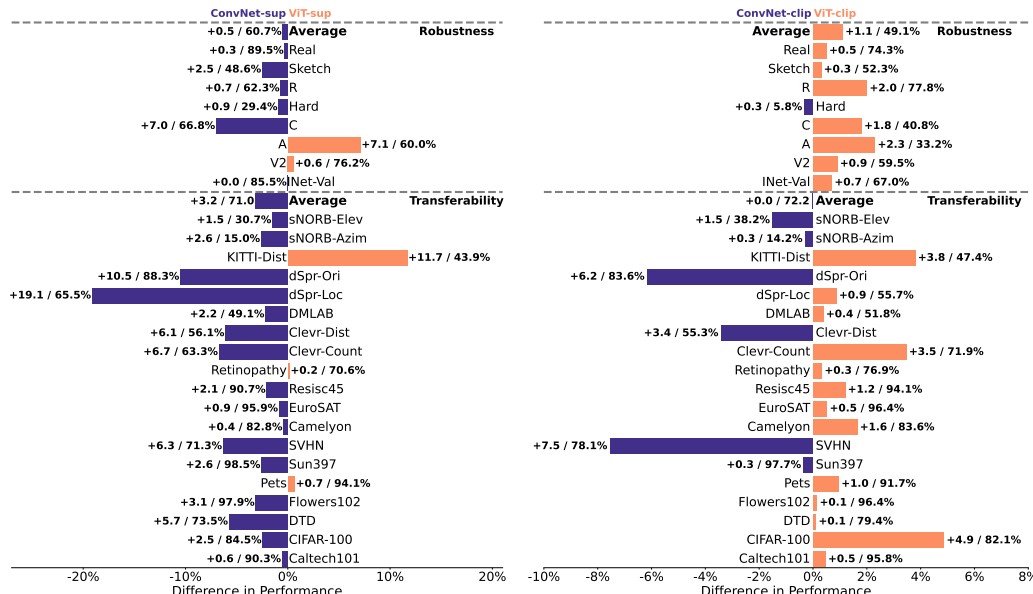

Figure 6: Results on robustness and transferability. CLIP models excel in transferability, while supervised models perform better on robustness benchmarks. Supervised ConvNeXt heavily dominates over supervised ViT.

To test the transferability of models, we adopted a VTAB benchmark (Zhai et al., 2019). It comprises 19 diverse datasets grouped into three subcategories: natural, specialized, and structured. We conduct a linear probing evaluation on frozen features, following the protocol from Ilharco et al. (2021). The results are shown Fig. 6 (bottom). Results grouped by subcategories are provided in Table 3 in Appendix.

We find that ConvNeXt strongly outperforms ViT for supervised models. For CLIP models, ViT and ConvNeXt demonstrate similar average performance, with many datasets showing a performance gap of less than 1%. CLIP models generally show better transferability on all three subgroups of VTAB, which is different from the robustness experiments. Their superiority can be attributed to the larger and more diverse volume of pretraining data (Ramanujan et al., 2023).

## 7. Transformation Invariance

We explore the model's ability to still recognize the input data when it goes through semantic-preserving transformations. Achieving various types of invariance is desirable because it enables the network to generalize well across different but semantically similar inputs, thereby enhancing its robustness and predictive power. In previous literature (Azulay & Weiss, 2018; Zhang, 2019), it has been shown that the performance of neural networks can be highly unstable even under simple input transformations, such as shifting by a few pixels.

We conduct experiments to assess three types of invariance: scale, shift, and resolution. We analyze the model's accuracy trends on the ImageNet-1K validation set as a function of varying scale/shift magnitude and image resolution. In crop experiments, the image is resized according to a given scale factor, and then a central crop is taken. In shift experiments, we adjust the crop location in the original (non-resized) image space and then take a crop, shifting along the longer side of the image. In resolution experiments with the ViT model, we interpolate positional embeddings to match the applied resolution.

The results are presented in Fig. 7. We observe a very consistent trend of ConvNeXt outperforming ViT under supervised training. This trend is reversed for CLIP models. Overall, models are reliable to shift transformation and less robust to scale and resolution transforms. For practical applications requiring high robustness to scale, shift and resolution transforms, our results indicate that ConvNeXt under supervised training could be the best choice.

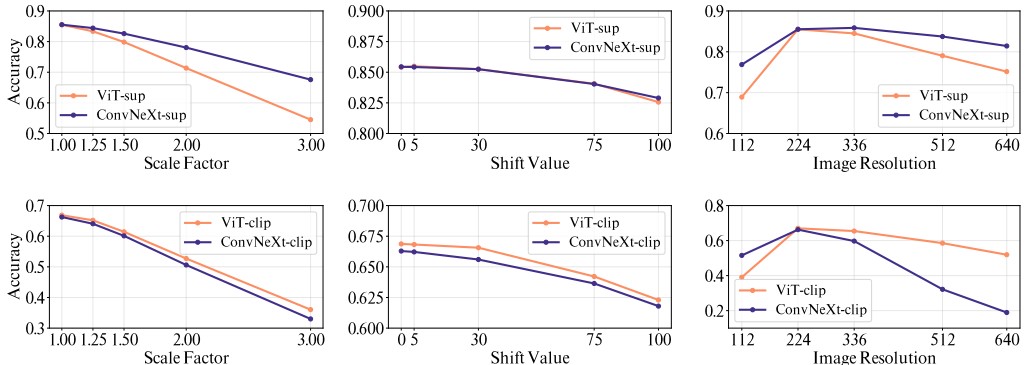

Figure 7: Scale, shift, and resolution invariance experiments on ImageNet-1K validation. ConvNeXt is better than ViT under supervised training. This trend is reversed for CLIP models.

## 8. Representation Similarity

Many metrics used to evaluate neural networks focus solely on the output, offering little insight into what shapes the final representation. To better understand these shaping factors, it is essential to "look under the hood" of the model. To facilitate this deeper analysis, representation similarity is often used as a metric. It quantifies the similarity between features produced by different layers or models, offering a fine-grained view of how data is processed and represented within the network.

In this section, we examine representation similarity using Centered Kernel Alignment (CKA) (Kornblith et al., 2019a). Specifically, we use a minibatch CKA (Nguyen et al., 2020), defined in Eq. 2:

$$\text{CKA}_{\text{minibatch}} = \frac{\frac{1}{k} \sum_{i=1}^{k} \text{HSIC}_1 \left( \mathbf{X}_i \mathbf{X}_i^\top, \mathbf{Y}_i \mathbf{Y}_i^\top \right)}{\sqrt{\frac{1}{k} \sum_{i=1}^{k} \text{HSIC}_1 \left( \mathbf{X}_i \mathbf{X}_i^\top, \mathbf{X}_i \mathbf{X}_i^\top \right)} \sqrt{\frac{1}{k} \sum_{i=1}^{k} \text{HSIC}_1 \left( \mathbf{Y}_i \mathbf{Y}_i^\top, \mathbf{Y}_i \mathbf{Y}_i^\top \right)}}, \quad (2)$$

where $\text{HSIC}_1$ is the unbiased estimator Hilbert Schmidt Independence Criteria (Gretton et al., 2007). This formulation of CKA measures the alignment or similarity between two high-dimensional feature activation matrices $\mathbf{X_i}$ and $\mathbf{Y_i}$ obtained from different layers for minibatch $i$. The metric employs HSIC as a building block to capture the dependence between two sets of features. To calculate minibatch CKA, we use 10% of images from ImageNet validation where all classes are sampled with a 10% rate. CKA scores range from 0 to 1, where higher values indicate greater similarity.

We compute a matrix of CKA values by comparing each pair of layers within the same model and layer type, i.e., attention/convolutional, linear, and normalization layers. For linear layers in ViT, we only consider layers from the MLPs blocks. For attention layers, we take the final representation from the self-attention blocks. We show the representation similarity matrices in Fig. 8.

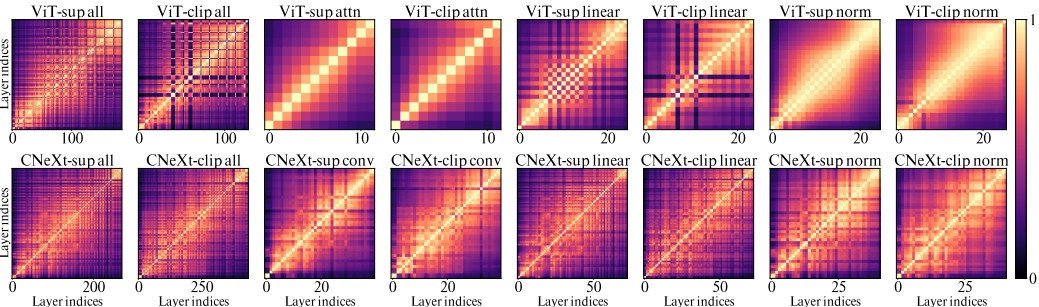

Figure 8: CKA maps for representations from the whole model (all), attention/conv, linear, and normalization layers for each model. ViT layers have different patterns, while in ConvNeXt, all the layers adhere to a consistent structure.

The CKA results serve as formal evidence corroborating what has been observed in prior experiments: there are marked differences in the representations across different architectures as well as between different training paradigms. All three types of ViT layers have different representation patterns compared to ConvNeXt. ViT's linear layers show a large difference in supervised and

CLIP models, driving the difference of representations. In the supervised ViT model, a pronounced "checkerboard" structure is visible in the layers from the 3rd to the 8th block. In ViT CLIP, no such phenomenon exists; however, there are two layers with representations highly dissimilar to those in any other layer of the network. Within ViT, the normalization layers stand out for their representational consistency, maintaining similarity across different blocks of the network.

ConvNeXt models follow very different representation patterns from ViTs. In ViTs, a drastic difference can be observed between all three layer types, unlike ConvNeXts, where conv, linear, and normalization layers follow similar fine-grained checkerboard structures. This comparison clearly shows the influence of architecture designs on learned representations.

## 4 RELATED WORK

**Architecture analysis.** In Raghu et al. (2021), the authors compared internal representations of ViTs and ConvNeXts and noted significant differences driven by the use of self-attention. Similarly, Ruiz et al. (2022) focused on comparing the performance of ViTs and ConvNeXt on synthetic data. Zhou et al. (2021) investigated the feature transferability of ConvNet and Transformers. Naseer et al. (2021) explored several interesting properties of Vision Transformers. Nguyen et al. (2020) studied the impact of neural network width and depth on learned representations.

**Training objective analysis.** A comprehensive analysis was conducted by Walmer et al. (2023), comparing ViTs trained with supervised, self-supervised, and CLIP objectives. Their study demonstrated the intricate internal behaviors of ViTs, especially in their attention mechanisms and learned representations. Grigg et al. (2021) and Gwilliam & Shrivastava (2022) analyzed the representations of models trained with supervised and self-supervised objectives, aiming to find similarities and differences between the two. Two recent works (Park et al., 2023; Shekhar et al., 2023) focused on investigating the effect of training objective in masked image modeling and contrastive learning.

**Limitations of ImageNet.** Several works (Beyer et al., 2020; Recht et al., 2019; Tsipras et al., 2020) highlight issues with the reliability and quality of ImageNet labels, suggesting they may not be good indicators of a model's ability to generalize. Studies by Kornblith et al. (2019b) and Miller et al. (2021) show a strong relationship between performance on ImageNet and other datasets, although this can depend on the model's architecture and training methods. Recent works by Richards et al. (2023) and Fang et al. (2023) emphasize that achieving high performance on ImageNet is not a guaranteed indicator of strong performance on more diverse or real-world datasets.

## 5 DISCUSSION

This study examined ConvNets and Transformers with supervised and CLIP training from multiple perspectives beyond the standard ImageNet accuracy metric. We found that each model can have its own distinct strengths. This suggests that model selection metric should depend on the target use cases, as standard performance measures may overlook key task-specific nuances.

**ConvNet vs Transformer.** In our analysis, we observed that often supervised ConvNeXt outperforms ViT, while CLIP-based ViT generally surpasses ConvNeXt. Three conclusions apply to both supervised and CLIP training: (1) ViT exhibits a greater shape bias. (2) ConvNeXt is calibrated no worse than ViT. (3) ConvNeXt excels over ViT on synthetic data. Moreover, representation similarity analysis exposes significant layer-wise differences between ConvNeXt and ViT. These observations show that architecture remains a pivotal factor in shaping representation learning.

**Supervised vs CLIP.** Based on our analysis, we arrive at the following key observations: (1) CLIP models suffer from overconfidence and are worse calibrated than supervised on both in and out-of-distribution data. (2) Supervised models perform better on robustness benchmarks, but likely because these benchmarks are ImageNet variants. This calls for new robustness benchmarks that are not related to ImageNet. (3) Conversely, CLIP models demonstrate significantly better feature transfer performance, likely attributed to their more diverse training data. We thus recommend using supervised models for ImageNet-like distributions and CLIP models for those with significant distribution shifts. (4) CLIP models make fewer mistakes relative to their ImageNet-1K accuracy than supervised. A decision between supervised and CLIP models is best based on target use cases.

**Reproducibility statement.** The code and instructions for running the experiments described in this paper are publicly available at this anonymous link: https://github.com/beyond-imagenet-accuracy/beyond-imagenet-accuracy.

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

# APPENDIX

## A ROBUSTNESS & TRANSFERABILITY RESULTS

In addition to the bar plot in Fig. 6, we also provide results in table form covering robustness benchmarks (Section 3.5) and transferability across VTAB subgroups (Section 3.6) below.

| Model | INet-Val | V2 | A | C | Hard | R | Sketch | Real | Average |
|---|---|---|---|---|---|---|---|---|---|
| ViT-sup | 85.5 | 76.2 | 60.0 | 59.8 | 28.5 | 61.6 | 46.1 | 89.2 | 60.2 |
| ConvNeXt-sup | 85.5 | 75.6 | 52.9 | 66.8 | 29.4 | 62.3 | 48.6 | 89.5 | 60.7 |
| ViT-clip | 67.0 | 59.5 | 33.2 | 40.8 | 5.5 | 77.8 | 52.3 | 74.3 | 49.1 |
| ConvNeXt-clip | 66.3 | 58.6 | 30.9 | 39.0 | 5.8 | 75.8 | 52.0 | 73.8 | 48.0 |

Table 2: Full results on robustness benchmarks.

| Model | Natural | Specialized | Structured | Overall |
|---|---|---|---|---|
| ViT-sup | 84.2 | 84.2 | 45.4 | 67.8 |
| ConvNeXt-sup | 87.1 | 85.0 | 50.0 | 71.0 |
| ViT-clip | 87.6 | 87.8 | 50.9 | 72.2 |
| ConvNeXt-clip | 87.8 | 86.9 | 51.2 | 72.2 |

Table 3: Transferability results on VTAB in subgroups. CLIP models are better on each of the dataset subgroups. Within each training method, ViT and ConvNeXt yield similar accuracy.

