# OpenReview forum: "ConvNet vs Transformer, Supervised vs CLIP: Beyond ImageNet Accuracy"
_ICLR.cc/2024/Conference — ICLR 2024 Conference Withdrawn Submission_

### Official Review · Reviewer_qUP4 · 2023-10-30

**Soundness:** 2 fair
**Presentation:** 2 fair
**Contribution:** 2 fair
**Rating:** 5
**Confidence:** 4

**Summary:**

This paper provides comparative analysis between ConvNeXt and ViT, and supervised and CLIP training. Various analyses show that they have their own pros and cons. For example, ViTs are shape-biased, the calibration of ConvNeXt is comparable with that of ViTs, and ConvNeXt outperforms ViTs on synthetic data. In terms of pre-training method, CLIP-pretrained neural networks are overconfident, supervised models are robust, CLIP models are transferrable, and CLIP models make fewer mistakes.

**Strengths:**

1. This paper is well-written and easy to follow. Even though most evaluation techniques are from previous works, I do not think that proposing a new evaluation method is not necessary to deliver a novel takeaway.
2. In the midst of a scarcity of analysis papers on ViTs and pre-trained neural networks, this paper is one of the few that offer such a comparative analysis. I believe there is value in this kind of analysis paper, especially when building intuition for developing new methods or selecting appropriate ones based on various observations.
3. Similarly, this paper provides a wide range of observations. In particular, I appreciate the observation that “CLIP models are overconfident” and “ConvNext outperforms ViT in terms of transferability”, which are novel observations for me.

**Weaknesses:**

1. One of the primary weaknesses of this paper is its organization. While the paper offers various straightforward observations, they are merely presented side by side. This gives the impression of a technical report rather than an academic paper. Building connections between sections might enhance the paper's coherence and demonstrate insights.
2. Some findings have already been demonstrated by previous research. For instance, [1] showed that ViTs are shape-biased. As pointed out by [2], this can be attributed to the finding that convolutions act as high-pass filters, whereas self-attentions function as low-pass filters. It's worth noting that while this paper suggests that ConvNeXt is more inclined to learn local features due to the inherent local nature of convolution operations, ConvNeXt does utilize convolutions with large kernel sizes, which focus not only local area but large area. Furthermore, [3-8] have demonstrated that CLIP-like models are highly generalizable. Additionally, the impact of other observations in this paper seems limited. Providing justification for such impact could enhance the paper's value.
3. I believe some of the approaches and conclusions are not adequately supported. For instance:
    1. I'm not fully convinced that comparing 'CLIP-pretrained models' with 'IN21K-pretrained and IN1K-finetuned models' provides a fair comparison. Although there might not be a significant difference, comparing 'CLIP-pretrained models' directly with 'IN21K-pretrained models' could be more equitable.
    2. The statement, "Training data and objectives could be more influential factors (than architectures)" raises concerns. I contend that the behaviors observed in Figure 4 are primarily because both ConvNeXt and ViT are pre-trained on IN21K, a large dataset. Similarly, the assertion that "supervised ConvNeXt has lower ECE scores" is debatable.
    3. I am not entirely convinced that using linear probing is the most suitable method to investigate transferability. Although it is one of the standard methods, I believe that fine-tuning the later layers of ViTs enhances their transferability. This suggests that a linear probing experiment might deviate from real-world scenarios, and incorporating experiments with (even partial) fine-tuning could strengthen the paper.
    4. In Fig 7, the trends for CLIP and supervised pre-trained models appear to be reversed. Delving deeper with explanations or analyses could be beneficial for readers.
    5. Section 3.8 seems somewhat speculative, and its main message is not readily clear. Regarding representation similarity, I'm skeptical about making a direct comparison. ConvNeXt employs various intricate components, such as complex module designs and multi-staging, making it challenging to draw an apple-to-apple comparison.

Overall, I am currently inclined towards rejection, though I am sitting on the fence. I will revisit this paper during the rebuttal period to finalize my assessment.

*Minor*: This paper posits that “supervised models tend to be slightly underconfident.” However, in terms of terminology, I believe the term ‘underconfident’ typically indicates a high ECE in an opposite way of 'overconfident'.

---

[1] Naseer, Muhammad Muzammal, et al. "Intriguing properties of vision transformers." Advances in Neural Information Processing Systems 34 (2021): 23296-23308.

[2] Park, Namuk, and Songkuk Kim. "How do vision transformers work?." arXiv preprint arXiv:2202.06709 (2022).

[3] Alec Radford, Jong Wook Kim, Chris Hallacy, Aditya Ramesh, Gabriel Goh, Sandhini Agarwal, Girish Sastry, Amanda Askell, Pamela Mishkin, Jack Clark, et al. Learning transferable visual models from natural language supervision. In International conference on machine learning, pp. 8748–8763. PMLR, 2021.

[4] Chao Jia, Yinfei Yang, Ye Xia, Yi-Ting Chen, Zarana Parekh, Hieu Pham, Quoc Le, Yun-Hsuan Sung, Zhen Li, and Tom Duerig. Scaling up visual and vision-language representation learning with noisy text supervision. In International Conference on Machine Learning, pp. 4904–4916. PMLR, 2021.

[5] Hieu Pham, Zihang Dai, Golnaz Ghiasi, Kenji Kawaguchi, Hanxiao Liu, Adams Wei Yu, Jiahui Yu, Yi-Ting Chen, Minh-Thang Luong, Yonghui Wu, et al. Combined scaling for zero-shot transfer learning. arXiv preprint arXiv:2111.10050, 2021.

[6] Alex Fang, Gabriel Ilharco, Mitchell Wortsman, Yuhao Wan, Vaishaal Shankar, Achal Dave, and Ludwig Schmidt. Data determines distributional robustness in contrastive language image pretraining (clip). In International Conference on Machine Learning, pp. 6216–6234. PMLR, 2022.

[7] Thao Nguyen, Gabriel Ilharco, Mitchell Wortsman, Sewoong Oh, and Ludwig Schmidt. Quality not quantity: On the interaction between dataset design and robustness of clip. In Advances in Neural Information Processing Systems, 2022.

[8] Shu, Yang, et al. "CLIPood: Generalizing CLIP to Out-of-Distributions." *arXiv preprint arXiv:2302.00864* (2023).

**Questions:**

Please refer to the weaknesses section above.

---

### Official Review · Reviewer_ZNBZ · 2023-10-31

**Soundness:** 3 good
**Presentation:** 4 excellent
**Contribution:** 2 fair
**Rating:** 5
**Confidence:** 3

**Summary:**

This paper compares two image classification architectures ConvNeXt and Vision Transformers with two learning objectives supervised and CLIP. It analyzes their failure modes, calibration, representation similarity, shape/texture bias, robustness and transferability. The justification for this study is very close accuracy for SOTA models that doesn't provide a lot of information about their performance in difference scenarios.

**Strengths:**

The paper is very well-written, the experiments are detailed and thorough, they generally support the paper's conclusions. Experimental results are clearly explained. In general, I like the idea of going beyond accuracy to claim "this model is better than that one".

**Weaknesses:**

The paper is very empirical, lacks any kind of theoretical contribution. I don't think the paper has a lot of significance because the main question is simple and the paper doesn't provide any explanations, merely reports some differences.

Minor nitpicks:
ImageNet-R isn't referenced the first time it is mentioned.
The last column in table 1 should indicate better it is a val accuracy, the current col title is confusing.

**Questions:**

While it is interesting to see the comparison between different architectures and objectives, it is unclear what conclusion a reader is supposed to make. The paper basically doesn't answer any "why" questions nor gives many practical takeaways.

Can any of observed differences be attributed to the training regime that is obviously different for two very different architectures?

Intermediate layer representations are more dissimilar in ViT. So... is it because ViT has attention layers and the other one has plenty of skip connections? Is it bad thing? In what context I may prefer more consistent layer representations?

CLIP models are more transferable? Yes, but they also show much lower performance on ImageNet. At what point the tradeoff starts to make sense?

On transformation invariance:
"We observe a very consistent trend of ConvNeXt outperforming ViT under supervised training. This trend is reversed for CLIP models".
Why is that? Why is that reversed for CLIP?

Basically none of the authors' observations and experiments are accompanied by ideas or explanations. It reminds more of a blog post.

---

### Official Review · Reviewer_vcFp · 2023-10-31

**Soundness:** 2 fair
**Presentation:** 3 good
**Contribution:** 2 fair
**Rating:** 3
**Confidence:** 4

**Summary:**

The paper claims that modern neural network architectures in vision are compared mainly on their performance on ImageNet. However, depending on the downstream application other metrics are desirable (For example, robustness to occlusions). The authors compare two architectures (DeiT-3 and ConvNext-B) trained on two regimes (ImageNet-supervised and CLIP-style training) on 8 different tasks.

The major insights from the paper are as follows:

### CLIP vs supervised models.
---------

* On ImageNet-X CLIP models have a better error ratio than supervised models, while on robustness and pug-Imagenet, clip models have a lower accuracy (apart from ImageNet-sketch and ImageNet-R)
* CLIP models show better transferability, have more shape bias and higher ECE.

### Convnext vs ViT models:
------

* ConvNext is better than ViT on PUG-ImageNet. Supervised ConvNext outperforms supervised ViT on transferability and transformation invariance.
* ViT models have a better shape bias. CLIP ViT outperforms CLIP ConvNext on robustness and transformation invariance.
* Representations learnt by ConvNext and ViT models are different.
* On calibration, ConvNext models are slightly better than ViT models.
* Other results are mixed.

**Strengths:**

* Using downstream benchmarks other than a single task to assess different models is a timely topic.
* The authors have explored a number of tasks.
* The paper is very easy to follow and all data is shown thoroughly.

**Weaknesses:**

It is difficult to understand the positioning of the paper. Several of the cited works have similar insights and have done a larger exploration. Here are a few examples:

* The finding that representations learnt by convolutional models and vision transformers are different has been thoroughly explored in Raghu et al 2021
* The finding that ViT models have more shape bias has been explored in Naseer et al 2021
* The finding that Clip models have higher effective robustness and better linear probe performance has been explored in the original Clip paper. (See Fig 10 of https://arxiv.org/pdf/2103.00020.pdf)

The paper should positioning itself in the context of this related work (for example: is this the focus on strictly on the newest generation image models? If yes, what insights does this paper offer which is different to the works metioned above)

IMHO this is quite difficult to fix in a rebuttal.

**Questions:**

Other Questions:

* Figure 4 suggests that CLIP models are not well calibrated but Fig 19 in https://arxiv.org/pdf/2106.07998.pdf has the opposite results across a range of CLIP models. Why is this the case?
* In Table 1, authors control for flops, params and accuracy. I suggest that they control for number of training examples as well. CLIP models might perform better just because they are trained on 400x more data.
* Why is ImageNet-X and PUG-Imagenet kept separate from the other ImageNet robustness results? They can also be considered as out of distribution ImageNet.
* For the image resolution experiments, the authors can consider extending the x-axis lower than 112 to test robustness to low resolution images.

---

### Official Review · Reviewer_nmW5 · 2023-11-02

**Soundness:** 3 good
**Presentation:** 3 good
**Contribution:** 2 fair
**Rating:** 5
**Confidence:** 4

**Summary:**

This paper compares the performances of ViT and ConvNeXt models from supervised and CLIP training frameworks.
Despite their similar ImageNet accuracies and compute requirements, these models differ in terms of types of mistakes, output calibration, transferability, and feature invariance, among others.

**Strengths:**

This paper provides a comprehensive benchmark of ViT and ConvNeXt from both supervised and CLIP training framework, including the types of mistakes, synthetic data, model calibration, shape / texture bias, robustness, transferability, transformation invariance, and representation similarity.

**Weaknesses:**

It's unclear on how the model differences between ViT and ConvNeXT or training frameworks between supervised and CLIP contribute to the performance differences in some categories. Please check the questions below.

**Questions:**

1. How does older ConvNeXt architectures, such as ResNet, differ from the more modern ResNext? What impact do these model architecture differences impose on the confidence histogram in Fig. 4?
2. How to isolate the impact of the large / diverse image data from the text for CLIP-trained models?
3. How might the depth (number of layers) impact the similarity? Is it possible to compare ViT and ResNext at the same layer scale?
4. How would the large accuracy gap between supervised models and CLIP models influence the benchmark results?